# Survival in Kidney and Bladder Cancers in Four Nordic Countries through a Half Century

**DOI:** 10.3390/cancers15102782

**Published:** 2023-05-16

**Authors:** Filip Tichanek, Asta Försti, Akseli Hemminki, Otto Hemminki, Kari Hemminki

**Affiliations:** 1Biomedical Center, Faculty of Medicine in Pilsen, Charles University, 30605 Pilsen, Czech Republic; f.tichanik@gmail.com; 2Institute of Pathological Physiology, Faculty of Medicine in Pilsen, Charles University, 32300 Pilsen, Czech Republic; 3Hopp Children’s Cancer Center (KiTZ), 69120 Heidelberg, Germany; 4Division of Pediatric Neurooncology, German Cancer Research Center (DKFZ), German Cancer Consortium (DKTK), 69120 Heidelberg, Germany; 5Cancer Gene Therapy Group, Translational Immunology Research Program, University of Helsinki, 00290 Helsinki, Finland; 6Comprehensive Cancer Center, Helsinki University Hospital, 00290 Helsinki, Finland; 7Department of Urology, Helsinki University Hospital, 00290 Helsinki, Finland; 8Division of Cancer Epidemiology, German Cancer Research Center (DKFZ), Im Neuenheimer Feld 580, 69120 Heidelberg, Germany

**Keywords:** uro-oncology, relative survival, conditional survival, treatment, surgery

## Abstract

**Simple Summary:**

Cancers in the urinary bladder and kidney are common in men and rarer in women. Cigarette smoking is a shared risk factor for both of these cancers. Some 50 years ago, survival in these cancers was low, and it was worse for kidney than bladder cancer. In the present study, we could show improvement in survival for these cancers in the Nordic countries, and similar improvements have also been observed in other countries. Kidney cancer survival improved remarkably well, as 50 years ago, only 20–30% of the patients survived 5 years, but currently, some 75% survive 5 years. In male bladder cancer, 5-year survival is still somewhat better than survival in kidney cancer, but female kidney cancer survival has caught up with that of bladder cancer. The reasons for this positive development for both of these cancers is earlier diagnosis as patients with blood in urine are readily taken for examinations. Additionally, treatment has become more efficient, and continuously new medications are being introduced.

**Abstract:**

Kidney and bladder cancers share etiology and relatively good recent survival, but long-term studies are rare. We analyzed survival for these cancers in Denmark, Finland, Norway (NO), and Sweden (SE) over a 50-year period (1971–2020). Relative 1- and 5-year survival data were obtained from the NORDCAN database, and we additionally calculated conditional 5/1-year survival. In 2016–2020, 5-year survivals for male kidney (79.0%) and bladder (81.6%) cancers were best in SE. For female kidney cancer, NO survival reached 80.0%, and for bladder cancer, SE survival reached 76.1%. The magnitude of 5-year survival improvements during the 50-year period in kidney cancer was over 40% units; for bladder cancer, the improvement was over 20% units. Survival in bladder cancer was worse for women than for men, particularly in year 1. In both cancers, deaths in the first year were approximately as many as in the subsequent 4 years. We could document an impressive development for kidney cancer with tripled male and doubled female 5-year survival in 50 years. Additionally, for bladder cancer, a steady improvement was recorded. The current challenges are to curb early mortality and target treatment to reduce long-term mortality.

## 1. Introduction

The incidence of renal cell carcinoma (RCC) is two- to fourfold higher in men than in women, and environmental risk factors include smoking, overweight, and obesity [1]. Family history is another risk factor, and von Hippel–Lindau gene mutations may be found in early-onset families [2,3]. Survival in RCC has generally improved due to early detection and improvements in treatment [1,4,5,6,7]. Novel imaging methods are able to detect increasingly small tumors, as has been noted in Sweden [8]. The standard treatment for RCC has been surgery, which has increasingly started to adopt minimally invasive techniques [8,9]. After 2006, kinase inhibitors have replaced cytokine treatments in metastatic kidney cancer, but their benefits for survival have remained debatable [5,10]. After 2015, multiple options for immunotherapy have become available [11,12].

Cancer of the urinary bladder accounts for more than 90% of urothelial cancers, while tumors of the renal pelvis and the ureter are rare entities [13]. In bladder cancer, male excess ranges from three- to fourfold, and its international incidence trends correlate with smoking prevalence [14]. Type 2 diabetes, occupational exposures, and family history are other risk factors [2,15]. Bladder and upper tract urothelial cancers are known components in Lynch syndrome [16,17]. In bladder cancer, nonmuscle invasive tumors, accounting close to 80% of all, are transurethrally resected, while muscle invasive tumors are typically treated with cystectomy; both treatments can be supplemented with chemotherapy or immunotherapy [18,19]. Nonmuscle invasive (superficial) tumors require surveillance, as there is a risk of later recurrence or progression [19]. Survival in bladder cancer has developed well, and the likely reasons are early detection, thanks to novel imaging technologies and better cystoscopy equipment, and improvements in treatment [18,20]. 

We will assess relative survival in kidney and bladder cancers in Denmark (DK), Finland (FI), Norway (NO), and Sweden (SE) from 1971 to 2020 with focus on changes in survival times. The study describes ‘real world’ status of cancer control in the Nordic countries, as patient access to care is guaranteed with minimal out-of-pocket costs. In addition to standard 1- and 5-year survival, we show data for conditional 5/1-year survival and annual changes in survival. Observed changes in country-specific survival rates are discussed in terms of the therapeutic and diagnostic landscape for these cancers. 

## 2. Methods

The data were obtained from the NORDCAN database 2.0 [21,22]. Iceland was not included in the analysis because of the small population. The database was accessed at the International Agency for Research on Cancer (IARC) website (https://nordcan.iarc.fr/en) (accessed in fall 2022) [23], and the available tools were used to extract data on incidence, mortality, and 1-year and 5-year survival. NORDCAN uses International Classification of Diseases (ICD) version 10 codes for cancer. RCC code C64 included also Wilms tumor patients, who accounted for less than 1.5% of all patients; however, for the age group below 50 years, they accounted for 12% of male and 20% of female patients. For bladder cancer, the codes were C65–68 (cancers of the pelvis, ureter, bladder), D09.0–1, D30.1–9, and D41.1–9 (in situ and tumors of undefined behavior at these sites).

Using NORDCAN, we extracted data of incidence, mortality, and 1-year and 5-year relative survival, and the follow-up was extended until death, emigration, or loss of follow-up or to the end of 2020. Incidence and mortality data were age-standardized for the world standard population. For incidence and mortality data, the starting date was 1961 (the earliest available for all countries), and the data were visualized using R (version 4.2.2) [24]. Survival data for relative survival were available from 1971 onwards, and the analysis was based on the cohort survival method for the first nine 5-year periods and a hybrid analysis combining period and cohort survival in the last period 2016–2020, as detailed [21]. In the hybrid analysis for the last 5-year period, data are used from the penultimate 5-year period to make up 5 years of survival. We used age-standardized relative survival, estimated by the NORDCAN team using the Pohar Perme estimator [25], a nonparametric approach to estimate relative survival. Age standardization was performed by weighting individual observations using external weights as defined at the IARC website. Age groups 0 to 89 were considered. The DK, FI, NO, and SE life tables were used to calculate the expected survival. 

Statistical modelling and data visualizations were performed using R statistical software (https://www.r-project.org), version 4.2.2, in the R studio environment (https://posit.co/) (code available at https://github.com/filip-tichanek/norde_kidney). For a graphic presentation of incidence and mortality rates, lines were smoothed by the cubic smoothing spline using the R function ‘smooth.spline’ with a smoothing parameter (‘spar’) of 0.4 and with 12 knots [26].

Time trends of 1-year and 5-year relative survival (in %; obtained from NORDCAN for each of the 5-year periods) were modelled using the Gaussian generalized additive models (GAM) with thin plate splines (5 knots) and identity links. GAM was fitted in a Bayesian framework using ‘brms’ R package (https://www.jstatsoft.org/article/view/v080i01) (version 2.18.0) as described [26]. For the 5/1-year survival ratio estimation, we divided the posterior draws from the 5-year survival model by the posterior draws from the 1-year model to obtain the posterior distribution of the conditional survival and its estimated annual changes over time. Thus, 5-year survival is an approximate product of 1-year and 5/1-year relative survivals. For all survival measures (relative 1-, 5/1-, and 5-year survival), we evaluated when the survival was changing over time with at least 95% plausibility (95% credible interval (Ci) of the first derivation of given survival measure did not cross zero for at least 5 years). We also aimed to identify ‘breakpoints’, i.e., times when the annual change of survival changed with at least 95% plausibility. This was assessed by the calculation of the second derivation of the given survival measure and its 95% Ci; the ‘breakpoint’ was defined as a peak value within at least a 3-year interval where 95% Ci for the second derivation did not cross zero. The whole procedure, including custom R functions and selection of breaking points, is shown in the commented R code at https://github.com/filip-tichanek/nord_kidney.

Comparisons with the US Surveillance, Epidemiology, and End Results (SEER) data for years 2012–2018 on non-Hispanic white women were performed through the National Cancer Institute’s Surveillance, Epidemiology, and End Results Program (https://seer.cancer.gov/statistics-network/explorer/application.html?site=1&data_type=1&graph_type=2&compareBy=sex&chk_sex_3=3&chk_sex_2=2&rate_type=2&race=1&age_range=1&hdn_stage=101&advopt_precision=1&advopt_show_ci=on&hdn_view=0&advopt_display=2#graphArea) (accessed in winter of 2023).

## 3. Results

### 3.1. Incidence and Mortality in the Nordic Countries

Age-standardized (world) incidence and mortality trends for kidney and bladder cancers for the period 2011 to 2020 are reported in Table 1. Male incidence rates for kidney cancer were around 10/100,000, and for bladder cancer, they were higher, particularly in DK and SE. Female rates for kidney cancer were about half of male rates and about a third of bladder cancer rates. Mortality rates were about 1/4 or less of the incidence rates. The related incidence and mortality curves between 1961 and 2020 are seen in Appendix A.

### 3.2. Relative Survival in the Nordic Countries

Periodic 1- and 5-year survival rates are shown from 1961 to 2020 in Appendix A, where significant periodic improvements are marked. Similarly, data for 5/1-year conditional survival are shown in Appendix A. 

Figure 1 shows relative 1-, 5/1-, and 5-year survival in DK men (a,b) and women (c,d) in kidney (a,c) and bladder (b,d) cancers. For male and female kidney cancer, improvement in survival came in two waves, interrupted by a slow improvement around 1990, followed by a positive trend before the year 2000; the increase was highest in 5-year survival, from over 40% in 2000 to almost 80% towards 2020. Towards the end, 1- and 5/1-year survival curves crossed each other. For bladder cancer, the increasing survival was almost linear, and the curves for 1- and 5/1-year survival were almost superimposable. Towards 2020, male 5-year survival exceeded 80%, almost 10% units over the female survival.

According to Figure 2, FI survival kinetics differed from those for DK. For kidney cancer, male increase in survival peaked late (1985) and almost stalled after the year 2000 but started to increase again towards the end; male 5-year survival reached over 70%. The increase in female kidney cancer survival slowed down, but the increasing trend was maintained, and 5-year survival ended at over 70%, as for men. For bladder cancer, the initial increase in survival curves slowed down, and after the year 2000, the increase was marginal. At the end, male 5-year survival was 80%, 5% units higher than female survival.

NO survival kinetics for both cancers were quite similar to the data for DK, although the starting levels were slightly higher than the DK levels (Figure 3). 

Survival in SE male kidney and bladder cancer improved in two waves, the first strong wave in the 1970s and the second weaker wave before (kidney cancer) or after (bladder cancer) 2010 (Figure 4). Towards the end, male and female 1- and 5/1-year survival curves for kidney cancer crossed each other. For female kidney and bladder cancers, survival increased throughout the 50 years. The final 5-year survival was over 80%, except for female bladder cancer, 75%. 

Appendix A lists survival in 5-year periods, and the country-specific survival can be compared. In 2016-2020, kidney cancer 5-year survival was best for SE men (79.0%) and worst for FI men (71.2%); for women, NO was on top (80.0%), and DK and FI shared the bottom rank (74.0%). For male bladder cancer, 5-year survival was best in SE (81.6%) and worst in FI (78.9%); for female survival, SE was on top (76.1%) and DK was worst (74.0%). Combined, SE had three top ranks and NO had one; FI was worst in two and DK in one, and one bottom rank was shared by DK and FI. For kidney cancer, the difference in 5-year survival was significant (nonoverlapping 95% CIs) between the best and the worst national survival; for bladder cancer, the difference was not significant. 

For kidney cancer, there was no sex difference in survival, but in bladder cancer, male survival was better, and the difference in 5-year survival in the final period was significant in all countries, except FI; the mean difference between the four countries was about 5.5% units. However, the sex difference almost disappeared in 5/1-year survival (1.5% units as the mean of the four countries, Appendix A).

Data from Figure 1, Figure 2, Figure 3 and Figure 4 and Appendix A allow the estimation of the magnitude of survival improvements over the 50-year period. The improvement in 5-year survival in male kidney cancer was 45% units, female kidney cancer about 40% units. For bladder cancer, the improvement was over 20% units. Improvement in kidney cancer was best in DK and in bladder cancer in FI among the Nordic countries. 

The US SEER 5-year survival in kidney cancer in 2012–2018 was 77.3% for men and 79.0% for women. For bladder cancer survival, figures from the SEER dataset were 78.9% for men and 74.8% for women.

## 4. Discussion

Here we report a significant improvement in cancer survival development in survival in kidney cancer, which tripled male 5-year survival in 50 years and doubled female survival. Compared with other solid cancers, kidney cancer has shown the largest improvements in survival among all cancers [7]. In the last period, 2016–2020, male kidney and bladder cancer 5-year survival was best in SE (79.0% and 81.6%). Smoking worsens survival in at least bladder and pelvic cancers, and SE men have had the lowest smoking levels in Europe [27,28,29]. The best survival for female kidney cancer was in NO (80.0%), and for bladder cancer, in SE (76.1%). Even though DK and FI could not match their Nordic partners in final survival rates, DK was able to increase kidney cancer and FI bladder cancer survival more than the other countries. FI presented a curious survival paradox for these cancers (bladder cancer at the level of the Nordic partners but kidney cancer well below them), both of which are treated at the urology clinic. While the lower rate of healthcare funding in FI might explain the differences, it is unclear why only kidney cancer shows this pattern [9,30]? In the 1980s, FI urologists established a FinnBladder collaboration to advance clinical research on bladder cancer, which may have offered clinical benefits for bladder over kidney cancer [31].

The US SEER 5-year survival in kidney cancer in 2012–2018 was 77.3% for men and 79.0% for women, somewhat below the best Nordic data of SE and NO, while the data of DK were slightly and the data of FI clearly inferior. For bladder cancer survival, figures from the SEER dataset were 78.9% for men and 74.8% for women. The US Female 5-year susvival rates were comparable to our Nordic data, while male Nordic values were clearly better, except in FI. Moreover, the US data for bladder cancer were without renal pelvic cancer, which was reported separately (51.3% for men and 48.9% for women; renal pelvic cancer was rare, only 3% of male and 8% of female bladder cancers). Of note, 5-year survival in the final 5-year period of our study, 2016–2020, is a combination of survival in that period and in the penultimate 5-year period due to the hybrid survival method applied (see Section 2). Thus, we cannot estimate the possible survival changes introduced towards the very end of the follow-up period. 

For kidney cancer in each Nordic country, 5-year survival improved somewhat more than 1-year survival, and thus, the curve for 5/1-year survival ran somewhat above the curve for 1-year survival (however, towards the end, 1-year curves surpassed 5/1-year curves). This implied that more people died within year 1 after diagnosis than in the 4 subsequent years. For NO, and particularly for DK, the rate of survival improvement was faster after the year 2000 than before it. In bladder cancer, the improvement over time was steady, except that in FI, the net improvement after the year 2000 was marginal. Recent improvements in cancer survival in DK may be ascribed to the national cancer policy that DK set up in the year 2000; this ensured funding for cancer care and brought about administrative changes for accelerated cancer care pathways [32].

For kidney cancer, the impressive increase in survival through the 50 years could not have been possible without means of reducing the proportion of metastatic tumors at diagnosis. The Swedish single-center data from the years between 1982 and 1993 showed that 44% of the newly diagnosed kidney cancer patients were metastatic and the mean tumor size was around 80 mm [33]. National data from the years 2005 to 2010 showed that the proportion of metastatic kidney cancers had dropped to 15% and the mean tumor size to 64 mm [34]. These data suggest that there has been a large shift in the proportion of early-stage cancers in two decades. The detection of early-stage tumors was correlated with the number of CT instruments in SE [35]. Probably more active surgery and laparoscopic and robot techniques have contributed as more patients can be considered for radical therapy as anesthesia risk is smaller. Alertness about kidney cancer in the primary care and population in general has increased. The FI experience in finding smaller tumors with improved imaging was similar to the SE one [36]. A DK study from the national kidney cancer register showed that survival improvements were weakest in patients with comorbidities [37]. An earlier study on kidney cancer survival reported age-dependent survival disadvantages, particularly among old women [9].

For kidney cancer, the most significant survival improvements in all countries, except FI, started at around the diagnostic year 1995, which was about 10 years before the introduction of the first tyrosine kinase inhibitor (2006). Many new tyrosine kinase approvals were subsequently granted, even though none of them showed prespecified overall survival benefit in the conducted trials [10]. Kidney cancer is an ‘immunogenic’ tumor where a spontaneous regression of even metastatic cancer has been proposed to take place in up to 1% of patients [38]. Similarly, immunotherapies are effective in kidney cancer, including high-dose interleukin 2 and checkpoint inhibitors, but the latter may be too recent to impact the data reported here (nivolumab was approved in 2015, followed by many others) [10]. 

A sex difference has been known to exist in bladder cancer survival, and many explanations have been brought forward [20]. In the present 5-year survival, the sex difference was 5.5% units for male advantage. However, when the sex difference was compared in 5/1-year survival, it had diminished to 1.5% units. Indeed, it has recently been reported that the sex difference disappears after the second year of diagnosis [39]. It has been suggested that the early sex difference could be due to delay in diagnosis [40]. Females are more used to blood in urine, the most common symptom of bladder cancer. Additionally, their first contact and examination by a gynecologist might delay the correct diagnosis typically needing cystoscopy in the urological consultation.

In SE, the standardized care pathway for bladder cancer patients was introduced in 2015 with the aim of reducing the time from the patient’s first symptom to his or her uniform management according to international guidelines [41]. Although the program improved many parameters, including time to transurethral resection of bladder tumors and the proportion of patients discussed at multidisciplinary team conferences, it is too early to judge its impact on survival. 

Macroscopic hematuria is the most common presenting symptom for bladder cancer, and the clinical guidelines recommend a follow-up investigation with cystoscopy and computed tomography urography, and for suspected bladder cancer, transurethral resection of the tumor [41]. An earlier SE study from 1997 to 2011 reported on clinical treatment and stage-specific survival, and concluded that relative survival did not improve despite more aggressive treatment [42]. The ‘more aggressive’ treatments included frequent second resections, increased use of intravesical Bacillus Calmette–Guerin or chemotherapy instillations, and introduction of neoadjuvant chemotherapies [18]. A previous study reported relatively poor bladder cancer survival in older patients [20]. 

The limitations in the present study are lacking pathological information of kidney and bladder cancers at diagnosis and any treatment information. To compensate, the advantages of the NORDCAN data are their uniquely long follow-up time from high-level cancer registries. It is not feasible to assume that comparable pathological data were available over 50 years, as it has turned out, for example, that even the closely collaborating Nordic cancer registries have difficulties in comparing data on tumor characteristics (stage) [43]. Lacking stage data does not allow the assessment of the contribution of early detection to increasing survival. However, comparison of 1- and 5/1-year survival allows the assessment of the death rates between the periods 0–1 and 1–5 years of diagnosis. These curves were close to each other for both kidney and bladder cancers, which is an achievement as no more patients were lost in 4 subsequent years than were lost in the first year. 

## 5. Conclusions

We could document the remarkably good improvement of 5-year survival in kidney cancer for which only 20–30% of the patients survived 5 years in 1971–1975 compared with some 75% survival in 2016–2020. Male 5-year survival in kidney cancer is still 5% units below that of bladder cancer, but female kidney cancer survival caught up with that of bladder cancer. The current guidelines for these cancers consider risk-adapted treatment with options for numerous types of immunotherapy [11,12].

## Figures and Tables

**Figure 1 cancers-15-02782-f001:**
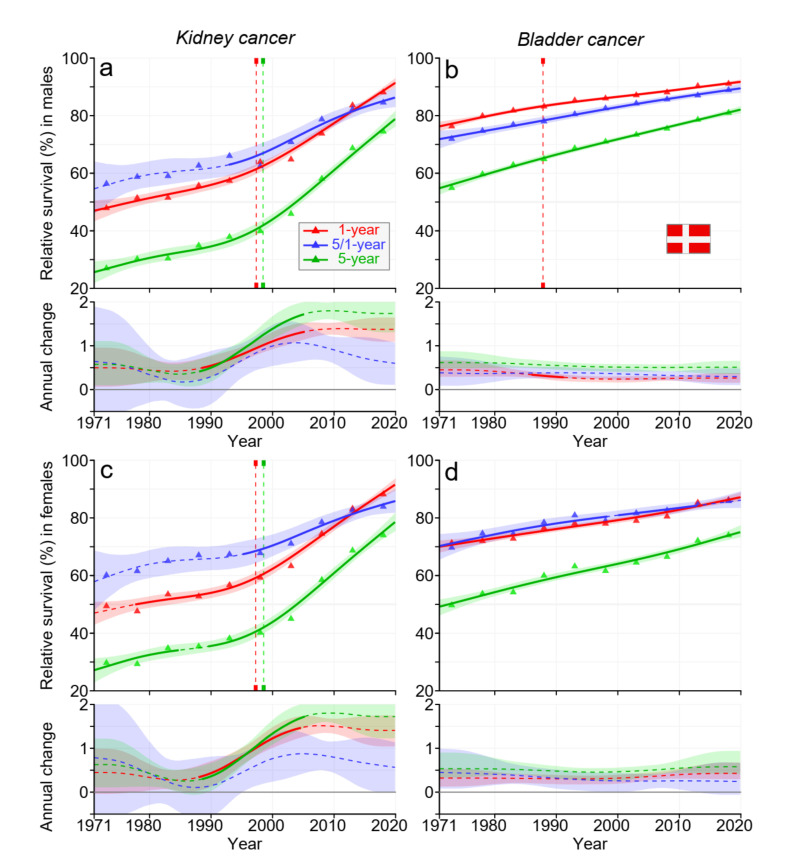
Relative 1-, 5/1-, and 5-year survival in Danish men (**a**,**b**) and women (**c**,**d**) in kidney (**a**,**c**) and bladder (**b**,**d**) cancers. The vertical lines show significant breakpoints in survival trends, and the bottom curves show estimated annual changes in survival. The solid line in survival curves and annual change curve indicate a plausible trend (see Section 2), whereas the dotted lines suggest lower plausibility. The shaded regions indicate 95% Bayesian credible intervals. All curves are color coded (see the insert).

**Figure 2 cancers-15-02782-f002:**
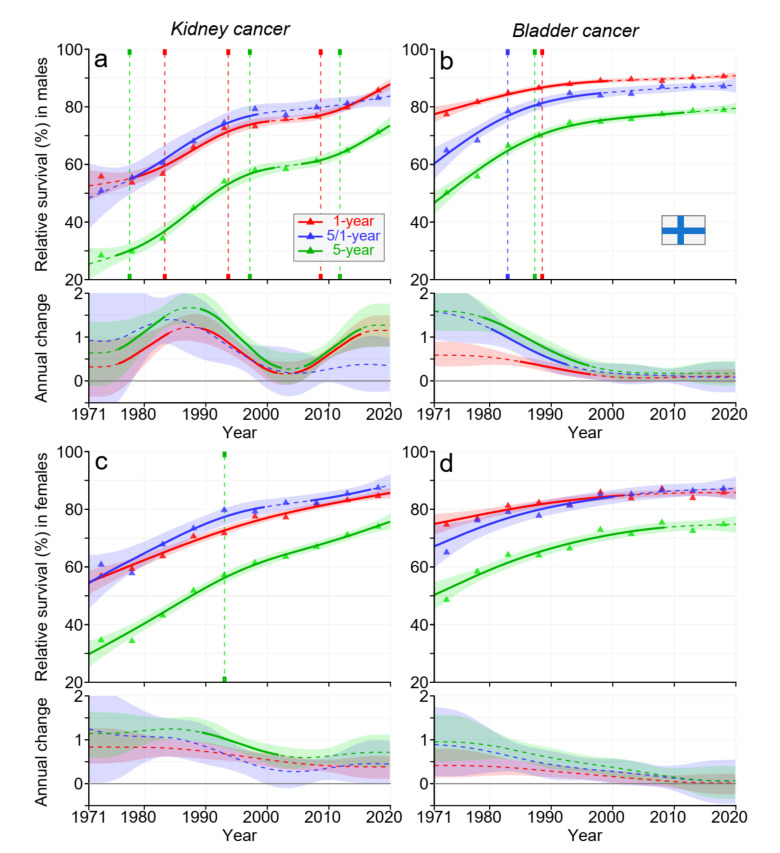
Relative 1-, 5/1-, and 5-year survival in Finnish men (**a**,**b**) and women (**c**,**d**) in kidney (**a**,**c**) and bladder (**b**,**d**) cancers. The vertical lines show significant breakpoints in survival trends, and the bottom curves show estimated annual changes in survival. The solid line in survival curves and annual change curve indicate a plausible trend (Section 2), whereas the dotted lines suggest otherwise. The shaded regions indicate 95% Bayesian credible intervals. All curves are color coded (see the insert).

**Figure 3 cancers-15-02782-f003:**
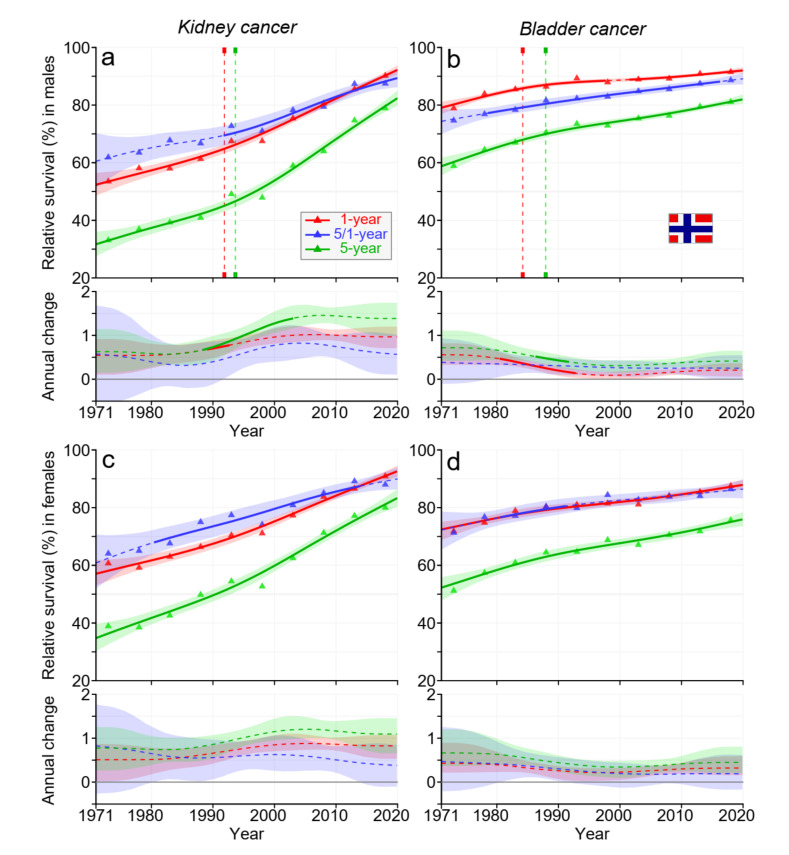
Relative 1-, 5/1-, and 5-year survival in Norwegian men (**a**,**b**) and women (**c**,**d**) in kidney (**a**,**c**) and bladder (**b**,**d**) cancers. The vertical lines show significant breakpoints in survival trends, and the bottom curves show estimated annual changes in survival. The solid line in survival curves and annual change curve indicate a plausible trend (Section 2), whereas the dotted lines suggest otherwise. The shaded regions indicate 95% Bayesian credible intervals. All curves are color coded (see the insert).

**Figure 4 cancers-15-02782-f004:**
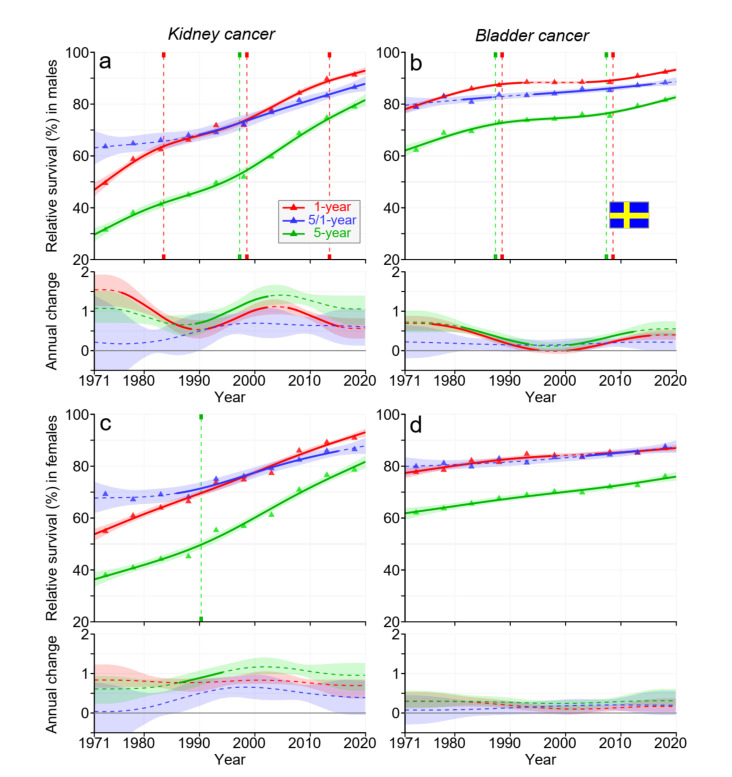
Relative 1-, 5/1-, and 5-year survival in Swedish men (**a**,**b**) and women (**c**,**d**) in kidney (**a**,**c**) and bladder (**b**,**d**) cancers. The vertical lines show significant breakpoints in survival trends, and the bottom curves show estimated annual changes in survival. The solid line in survival curves and annual change curve indicate a plausible trend (Section 2), whereas the dotted lines suggest otherwise. The shaded regions indicate 95% Bayesian credible intervals. All curves are color coded (see the insert).

**Table 1 cancers-15-02782-t001:** Incidence (A) and mortality (B) in kidney and bladder cancers from 2011 to 2020, separately for men (left part) and women (right part).

**(A) Case numbers, age-standardized incidence/100,000 (ASR), and cumulative risk**					
**Men**	**ASR (World)**	**Cum. Risk % (0–74)**	**Women**	**ASR (World)**	**Cum. Risk % (0–74)**
**Kidney**					
Denmark, 6306	12.2	1.5	Denmark, 3137	5.5	0.66
Finland, 5774	10.9	1.3	Finland, 3951	6.1	0.69
Norway, 5832	13.2	1.6	Norway, 2748	5.8	0.68
Sweden, 7748	8.5	1.0	Sweden, 4283	4.5	0.53
**Bladder**					
Denmark, 16,892	27.1	3.1	Denmark, 6076	8.8	1.1
Finland, 9849	15.4	1.7	Finland, 2972	3.6	0.42
Norway, 11,042	20.7	2.3	Norway, 3827	6.3	0.73
Sweden, 21,980	18.6	2.2	Sweden, 7775	6.0	0.72
**(B) Death numbers, mortality (ASR), and cumulative mortality**					
**Kidney**					
Denmark, 1793	3.0	0.35	Denmark, 1018	1.3	0.14
Finland, 2053	3.3	0.37	Finland, 1543	1.7	0.17
Norway, 1668	3.2	0.35	Norway, 891	1.3	0.13
Sweden, 3121	2.6	0.27	Sweden, 2061	1.3	0.12
**Bladder**					
Denmark, 3872	5.4	0.49	Denmark, 1809	2.0	0.19
Finland, 2257	3.1	0.26	Finland, 981	0.89	0.08
Norway, 2658	4.2	0.33	Norway, 1159	1.4	0.13
Sweden, 5751	4.0	0.34	Sweden, 2648	1.5	0.14

## Data Availability

Full statistical R code is available at https://github.com/filip-tichanek/nord_kidney.

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
