# Peer review of "Survival in Kidney and Bladder Cancers in Four Nordic Countries through a Half Century"

_cancers, 2023, doi:10.3390/cancers15102782_

Round 1
Reviewer 1 Report
The manuscript is well written and provides valuable information on the topic. Moreover, the paper reveals a huge potential of long-lasting population-based cancer registration in cancer survival research.
I have a few minor comments:
1. As there are five Nordic countries, please explain to the reader why Iceland was excluded from the analysis.
2. Currently, the title of the manuscript is misleading. It should read “… cancers in four Nordic countries …”.
3. Use consistently the words “male-female” or “men-women”, but do not use these words interchangeably (see text and graphs). The same concerns "data was" or "data were".
4. The readability of all line graphs can be considerably improved by replacing a gray background with a white one. Additionally, for better clarity, the y-axis of the bottom graphs may be labelled as “Annual change in survival (%)” instead of “δ (survival)”.
5. Also, set the y-axis of the relative survival graphs start at zero as suggested by experts. If visually better, break this axis between zero and 20. Write at the start of the x-axis of the relative survival graphs the number 1971.
6. Replace "counties" with "countries" (p. 10).
7. "A previous study reported age-group specific survival in bladder and the relative poor survival of the old patients" should probably read "A previous study reported relatively poor bladder cancer survival in the older patients".
8. Use "spelling and grammar" to check for typos.
9. IARC = International Agency for Research on Cancer, but not International Agency for Cancer (p. 3 and p. 11).
10. References: Abbreviate the title of the journal: ref. 5, 18, 33, 40, 41.
Reviewer 2 Report
This study analyzed survival for kidney and bladder cancers from Denmark, Finland, Norway, and Sweden over a 50-years period. The paper was well written. Comments below:
1. The authors said, “Survival analysis applied the cohort survival method for the first nine 5-year periods, and a hybrid analysis combining period and cohort survival in the last period 2016-2020 (21)”. This could be described in more detail, especially what “hybrid analysis” meant. A paper on the “survivorship-period-cohort model” could be helpful for the present purpose: “A survivorship–period–cohort model for cancer survival: application to liver cancer in Taiwan, 1997–2016. Am J Epidemiol 2021;190:1961-1968”.
2. The authors should provide more details about the age-standardization of survival data, the Pohar Perme estimator, and the Gaussian generalized additive models.
3. The 5-year survival is the product of the 1-year and 5/1-year survival. The authors can comment on that.
Round 2
Reviewer 2 Report
The ‘period approach’ was used to estimate the 5-year survival for the latest 5-year period. The author should discuss the disadvantage of the period approach. Also, using IARC data should not deter the authors from trying new methods other than those provided by IARC. The methodologies used in the paper, such as the age-standardization of survival data, the Pohar Perme estimator, and the Gaussian generalized additive model, were still inadequately described. The authors’ being the users rather than the developers of the methods is no excuse for not providing sufficient details of the methods used in their study. Providing R code and its output is certainly not enough.